# Microstructure and Properties of a Graphene Reinforced Cu–Cr–Mg Composite

**DOI:** 10.3390/ma15176166

**Published:** 2022-09-05

**Authors:** Ruiyu Lu, Bin Liu, Huichao Cheng, Shenghan Gao, Tiejun Li, Jia Li, Qihong Fang

**Affiliations:** 1State Key Lab of Powder Metallurgy, Central South University, Changsha 410083, China; 2State Key Lab of Advanced Design and Manufacturing for Vehicle Body, Hunan University, Changsha 410082, China

**Keywords:** Cu–Cr–Mg composite, graphene, layered structure, ball-milling

## Abstract

To improve the graphene/copper interfacial bonding and the strength of the copper matrix, Cu–Cr–Mg alloy powder and graphene nanosheets (GNPs) have been used as raw materials in the preparation of a layered graphene/Cu–Cr–Mg composite through high-energy ball-milling and fast hot-pressing sintering. The microstructure of the composite after sintering, as well as the effect of graphene on the mechanical properties and conductivity of the composite, are also studied. The results show that the tensile strength of the composite material reached a value of 349 MPa, which is 46% higher than that of the copper matrix, and the reinforcement efficiency of graphene is as large as 136. Furthermore, the electrical conductivity of the composite material was 81.6% IACS, which is only 0.90% IACS lower than that of the copper matrix. The Cr and Mg elements are found to diffuse to the interface of the graphene/copper composite during sintering, and finely dispersed chromium carbide particles are found to significantly improve the interfacial bonding strength of the composite. Thus, graphene could effectively improve the mechanical properties of the composite while maintaining a high electrical conductivity.

## 1. Introduction

Copper alloys are widely used in electrodes, rail transit, electronic packaging, and electrical contact. The most optimal properties of the copper alloys include high strength while providing excellent electrical and thermal conductivities. The fast development of the mechanical, electronic, and transportation industries has given rise to high expectations of the properties of copper alloys [1]. However, traditional copper alloys have been unable to meet these requirements. Consequently, copper matrix composites have come to attract a lot of attention in the scientific community. At present, the reinforcing components that are added to the copper composite are mainly oxide and carbide particles. It has been shown that the mechanical properties will then be significantly improved while the electrical conductivity is still reduced to a certain extent [2]. The reason is that the conductivity of these reinforcement materials is not high. In contrast, graphene has demonstrated the most excellent mechanical properties and electrical conductivity. Thus, a graphene/copper composite is expected to be the most promising copper alloy system for achieving concurrent high strength and high conductivity [3]. Shao et al. fabricated a graphene nanoplate/copper (GNPs/Cu) matrix composite by using flake powder metallurgy and spark plasma sintering (SPS). They found that by adding 0.2 wt.% GNPs to the Cu matrix, the tensile strength could be increased by 27% (i.e., to 233 MPa) [4]. Furthermore, Shu et al. prepared a high-quality graphene-reinforced copper-based composite by using a gel-assisted method, and the tensile strength of the composite was increased to 253 MPa [5]. Jiang et al. also prepared a high-quality graphene-reinforced copper-based composite by using electrostatic self-assembly. The yield strength of the composite increased from 95 MPa to 172 MPa, but the electrical conductivity of the composite decreased from 99.1% IACS to 84.2% IACS [6].

However, the experimental mechanical properties of graphene/copper composites are usually much lower than the theoretical expectations. This is mainly due to the uneven dispersion of graphene and the weak interfacial bonding between graphene and copper. At present, considerable results have been achieved for the dispersion of graphene in a Cu matrix. In contrast, there are few reports on the optimization of the graphene/copper interface in the composite. The decoration of a graphene surface with metal particles is, at present, a method that can be used to improve the binding within the graphene-copper interface. Since the modified graphene and copper matrix show good chemical affinities, the problem with poor interfacial wettability can be well improved. According to the current reports, Cu, Ni, and Ag can significantly enhance the interfacial binding and mechanical properties of the graphene/Cu composite [7,8,9]. Another approach is to increase the interfacial bond strength by using matrix alloying, in which carbide particles are formed in the interface during sintering. Dong et al. prepared a GNPs/CuW composite by using ball-milling of a mixture of copper powder, tungsten powder, and graphene nanosheets. The tensile strength increased to 295 MPa, which was 72.5% higher than that of GNPs/Cu [10]. Furthermore, Shi et al. prepared a GNPs/CuTi composite by vibrating a mixture of copper powder, titanium powder, and graphene nanosheets (GNPs). Compared with GNPs/Cu composites, the yield strength and tensile strength of the GNPs/CuTi composites were then increased by 115.9% and 66.7%, respectively [11]. Chu et al. prepared reduced graphene oxide (RGO)/CuCr composites by using pre-alloyed CuCr powders. The tensile strength of this type of composite was 19% higher than that of an RGO/Cu composite. At the same time, Chu et al. found that the size of the carbide particles in the interface of the composite material will significantly increase with an increase in temperature during the sintering process. However, this was found to negatively affect the mechanical properties of the composite material [12]. In addition, the current research on matrix alloying mainly focuses on the improvement of interfacial bonding, and there are few reports on strengthening the copper matrix [13].

In the present study, Cu–Cr–Mg alloy powder and graphene nanosheets were used as raw materials in the preparation of a GNPs/Cu–Cr–Mg composite. The method of high-energy ball-milling was used in this preparation. The problems presented above were then solved by including Cr and Mg elements in the alloy powder. The interfacial bond strength in the composite was improved by the addition of Cr, and the added Mg element did not only play a major role in the solid solution strengthening but also caused segregation in the areas where the Cr element was enriched (to control the size of the carbides).

## 2. Materials and Methods

### 2.1. Experimental Materials

The raw materials that were used in the experiments were a gas-atomized Cu–Cr–Mg alloy powder, graphene nanosheets (GNPs, with a purity of 99.5%, a sheet thickness of 3–10 nm, and a specific surface area of 31.657 m^2^/g), stearic acid (Analytical grade AR), absolute ethanol (Analytical grade AR), and argon (99.99%). Alloy powder was prepared by gas atomization method using high purity electrolytic copper (99.99%), cast Cu-10Cr (wt.%) intermediate alloy, and cast Cu-10Mg (wt.%) intermediate alloy as raw materials. The resulting powders were sieved, and the powders with average particle size less than 50 μm were collected. The chemical composition of the Cu–Cr–Mg alloy powder is listed in Table 1.

### 2.2. Material Preparation

Firstly, 0.2 g of GNPs were added to 250 mL of absolute ethanol. This mixture was ultrasonically blended for 30 min in the preparation of a uniformly dispersed graphene suspension. As the next step, 99.8 g of the Cu–Cr–Mg alloy powder was added to the suspension in an argon atmosphere. The mixture was ultrasonically dispersed for 30 min, then magnetically stirred at 80 °C for 30 min. It was, thereafter, vacuum dried at 60 °C for 12 h. The GNPs/Cu-Cr-Mg composite powder was then finally obtained.

In the next step, the GNPs/Cu–Cr–Mg composite powder was ball-milled for 6 h in an argon atmosphere, and 1 wt.% stearic was added as a process control agent. The ball-to-powder ratio was 10:1. The ball-milled powder was, thereafter, placed in a mold with a diameter of 40 mm. It was sintered at 900 °C and 50 MPa for 10 min with a 100 °C/min heating rate under vacuum conditions. The Cu–Cr–Mg matrix material was prepared by using the same method. 

The specific preparation process of the GNPs/Cu–Cr–Mg composites is demonstrated in Figure 1.

### 2.3. Characterization

The composite was sintered in fast hot-pressing sintering (FHP-828, HAATN, Jiangsu, China). Raman spectroscopy (Raman; LabRAM HR800, HORIBA Jobin Yvon, Paris, France) was used to analyze the structure of the GNPs and the composite at different stages by using a light source with a wavelength of 532 nm and a scanning rate in the range of 1000–3000 cm^−1^. The carbon content in the composite powder was analyzed by using a carbon-sulfur analyzer (CS-600, LECO, Champaign, IL, USA). Moreover, the density of the composite was measured by using the Archimedes drainage method (ET-320RP, ETNALN, Beijing, China). The phase composition of the composites was characterized by X-ray diffraction (XRD, D/Max 2550, Rigaku Corporation, Tokyo, Japan). The electrical conductivity of the composite was tested by using a digital eddy current metal conductometer (Sigma 2008, Tianyan, Fujian, China). Furthermore, field emission scanning electron microscopy (SEM; FEI Quanta FEG 250, FEI, Hillsboro, OH, USA) was used to analyze the microscopic morphology of the composite powder, as well as the microstructure and tensile fracture of the composite material. The surface elemental analysis of the prepared composite blocks was carried out by using an electron probe microanalyzer (EPMA; JXA-8530F, JEOL, Tokyo, Japan). Moreover, the composite was subjected to room temperature tensile tests by using a universal testing machine (Instron 3369, Norwood, MA, USA) with a strain rate of 1 mm/min.

## 3. Results and Discussion

### 3.1. Characterization of the GNPs/Cu–Cr–Mg Mixture Powder

Figure 2 shows the surface morphology of the raw powder and the GNPs/Cu–Cr–Mg mixture powder after ball milling. The raw alloy powders are mainly spherical and nearly spherical. The graphene nanosheet has complete sheet structure and smooth surface, and wrinkles and folds of the graphene can be observed obviously. The similarity in shape of the graphene and metal powder is a key factor in achieving a uniform dispersion of graphene and enhancing the GNPs/Cu–Cr–Mg interfacial binding force. Compared with a spherical powder, the two-dimensional structures of the flake powder and of graphene are very similar, which can effectively increase the contact area between the powder and graphene. This will not only enable improved support for the GNPs but also improve the bond strength between the alloy matrix and the GNPs [4,12,14,15]. As can be seen in Figure 2, due to the high rotation rate and high ball-to-powder ratio, the repeated compression impact force exerted by the ball on the powder during the ball-milling process is large, and the initially spherical Cu–Cr–Mg alloy powder particles have been flattened. That is, the powder particles have been crushed and melded together to form flakes with flat surfaces [16,17]. There was no obvious graphene agglomeration on these powder particle surfaces, which indicates that a uniform distribution of graphene has been achieved by using liquid-phase mixing and high-energy ball-milling.

### 3.2. Microstructure Characterization of GNPs/Cu–Cr–Mg Composites

To understand the microstructure of the sintered GNPs/Cu–Cr–Mg composites, the samples were wire-electrode cut in parallel with the sintering pressing direction. Figure 3a,b shows the microstructures that are in parallel with the pressing direction. The composite image consists of a light-colored alloy matrix and dark-colored graphene, with an obvious layered structure. The reason for this layered structure is that the flake powder particles have a large ratio of diameter-to-thickness and will, therefore, self-assemble into a layered structure under the action of gravity during the filling of the graphite mold. This also realizes the directional distribution of graphene in the material [4,18,19]. Moreover, a uniform distribution of graphene on the flake powder particles has resulted in a preserved layered structure of the composite material [3]. The distribution of graphene has been found to hinder the metallurgical bonding and the recrystallization between the different powder particles during hot-pressing sintering. Figure 3c,d show the microstructure of the Cu–Cr–Mg matrix that is in parallel with the pressing direction. It can be observed that metallurgical binding and recrystallization occurred between the flake powder particles during sintering when no graphene was added. The layered structure was, therefore, not preserved.

The XRD results of the composite are shown in Figure 4. Only the diffraction peak of Cu is detected, but the peaks of graphene, Cr, and Mg are not found, which is mainly because the amount of them added is too small to be detected by the XRD device.

### 3.3. Form and Distribution of Graphene

An EPMA analysis was performed to further study the distribution of graphene and alloying elements in the composite. The EPMA surface scan results can be seen in Figure 5. Figure 5c shows the distribution of C atoms in the composite material. It can be seen that C atoms are concentrated at the boundaries of the flake powder particles. A comparison of Figure 5a,b does also show that the C atoms are dispersed at the boundaries of graphene. Moreover, Figure 5d,e shows the distribution of Cr and Mg elements in the composite material. It is clear that also Cr and Mg have been mainly distributed at the boundaries of the flake powder particles. The graphene structure was destroyed during the ball-milling and dispersion processes, and amorphous carbon could be formed at the edges of the graphene. Some of the Cr atoms in the matrix could also diffuse to the powder particle boundaries during the sintering process. They reacted with the defective C atoms to form carbide particles at these graphene edges. The interfacial binding strength between graphene and the matrix was, thereby, improved [12]. In the process of diffusion of the Cr element, Mg atoms were discharged around it. With the progress of the reaction, Mg segregation could be formed around the carbide particles, which hindered Cr diffusion to the interface and further reaction with defective C atoms, thus limiting the size of carbide particles at the interface of composite materials.

The structures of pristine GNPs, as well as the structures of graphene before and after rapid hot-pressing sintering, were analyzed by using Raman spectroscopy. Three characteristic peaks of graphene can be seen in the Raman spectrum in Figure 6, namely, the D peak (~1340 cm^−1^), the G peak (~1580 cm^−1^), and the 2D peak (~2700 cm^−1^). The D peak is known as the defect band, and the 2D peak is known as the G0 peak. The D peak has been caused by the vibration of the defects in the graphene plane, or on the edge, which reveals the structural defects in graphene. Moreover, the G peak corresponds to the in-plane vibration of the sp^2^ hybridized carbon atoms. The intensity ratio of the D-to-G peak can be used to evaluate the defects and quality of graphene [6,20]. In addition to the enhancement of the D peak, the increase in defect density may also lead to a broadening of the D peak. However, the intensity ratio of the peaks can only reflect the change in D peak intensity. We have, therefore, used the area ratio of the D-to-G peak (I_D_/I_G_) to describe the change in graphene defect density, which has also been reported in other studies [10,21]. To accurately estimate this ratio, Gauss–Lorentz area numerical integration has been used to fit the D and G peaks. The I_D_/I_G_ of the pristine GNPs was found to be as small as 0.17, which indicates that the quality of graphene is high. However, the I_D_/I_G_ of graphene was found to increase to 0.45 after ball-milling, which indicates that the sp^2^ structure of graphene has been damaged to some extent. The reason is that in the process of high-energy ball-milling, the graphene and ZrO_2_ balls will collide with each other, causing some damage to the surface and edges of graphene [22]. Moreover, the I_D_/I_G_ value of graphene did slightly increase to 0.55 as a result of sintering. It can, in part, be explained by the plastic deformation that takes place in the powder during the sintering process. On the other hand, Cr in the alloy matrix can react with amorphous carbon at the edge of the graphene during the sintering process, which increases the number of defects in the graphene [11,23,24].

### 3.4. Influence by Graphene on the Properties of the Composites

The relative density and electrical conductivity of the composites are listed in Table 2. After sintering, the relative density of the two materials reached more than 99.0%, and the relative density of the composite material decreased by 0.4% compared with the matrix. Compared with other reports, the decline was small [22,25,26]. Due to the poor interface wettability between graphene and copper, there are many micropores and cracks at the graphene/copper interface after sintering, resulting in a significant decrease in the density of the composite [7]. Carbide-forming elements such as Cr and W can eliminate the defects at the interface and improve the density of composites by forming carbides [10,27]. Therefore, due to the addition of Cr and Mg, the disseminated carbide particles are formed on the interface of the composite, which improves the interface bonding and avoids the large decrease in the density.

The conductivity of the Cu–Cr–Mg matrix material was found to have a value of 82.4% IACS. Moreover, the conductivity of the GNPs/Cu–Cr–Mg composite had a value of 81.5% IACS, which is only 0.9% IACS lower than that of the matrix. This decrease in conductivity is small compared with other studies [6,8,25]. The addition of a reinforcing phase will usually reduce the electrical conductivity of a copper matrix composite. This is mainly due to an extra interface between the metal matrix and the reinforcement phase, which scatters the electrons during the electron transport process. It results in a decrease in the mean-free path (MFP) of the electrons [26,28]. For graphene/copper composites, the structural integrity of graphene and the interfacial bonding strength of graphene/copper are important factors affecting the electrical conductivity of the composites. The higher the structural integrity graphene has, the better excellent electrical conductivity graphene can retain, which causes a smaller impact on the electrical conductivity of the composite. The higher bonding strength of the graphene/copper interface has the better wettability of the graphene/copper interface, which reduces the scattering of electrons from the interface [29]. Therefore, there are two main reasons for the small decrease in conductivity. On one hand, graphene is prepared by using graphite embedding and stripping, which does not involve the oxidation-reduction process. Thus, the conjugated carbon network in graphite can be well preserved in graphene, ensuring its excellent conductivity. On the other hand, it benefits from the nanocarbide particles that are formed at the interface of the graphene/copper composite. The wettability between graphene and copper is improved, which greatly reduces the interfacial resistance of the graphene/copper and decreases the range of conductivity [6,25,29,30].

The tensile properties of the GNPs/Cu–Cr–Mg composite are shown in Figure 7. Figure 7a shows the engineering stress-strain curve of the matrix material and the composite, and Figure 7b shows the tensile strength (TS), yield strength (YS), and elongation (EL) of the matrix material and the composite. The yield strength and tensile strength of the matrix material have been found to be 118 MPa and 238 MPa, respectively. Moreover, the yield strength and tensile strength of the composite were found to be 280 MPa and 349 MPa, respectively, which are 137% and 47%, respectively, higher than that of the matrix material. Moreover, graphene was found to be uniformly distributed in the GNPs/Cu–Cr–Mg composite. In addition, carbide particles precipitated at the graphene/copper interface, which provides a strong interfacial binding and an enhanced load transfer capacity of the composite [12,31,32]. However, the elongation of the composite was found to drop from 13.1% to 7.6%. This was mainly due to the low pressure of 50 MPa in the sintering process, which caused too low a density in the composite. Moreover, there were still some small voids between the graphene and the matrix. These voids can become the source of cracks in the tensile process, which will seriously reduce the plasticity of the composite. Moreover, the brittle carbides at the interface of the composites can also have a certain effect on the elongation of the composites [15].

The strengthening efficiency (R) of a reinforced phase is an important parameter in the evaluation of the strengthening effect of a reinforced phase in the composite. It is defined as presented in Equation (1) as follows:
(1)$$R=\frac{\left(\sigma_{c}-\sigma_{m}\right)}{V_{f}\cdot \sigma_{m}}$$
where σ_c_ and σ_m_ are the yield strengths of the composite and the matrix, respectively, and V_f_ is the volume fraction of graphene. Figure 7c shows the strengthening efficiency of various methods for the copper matrix composite. The strengthening efficiency of graphene has been found to be 136, which indicates that the strengthening efficiency in the present study is better than the efficiency reported in other studies [13,24,33,34,35].

Furthermore, Figure 8a shows the tensile fracture of the matrix material. Since there is no graphene barrier between the flake powder particles, the material has recrystallized during the sintering process. Thus, there is no obvious layered structure at the tensile fracture. Moreover, Figure 8b shows the tensile fracture of the GNPs/Cu–Cr–Mg composite. It can be seen that the metallurgical bonding has been decreased, which is due to the presence of a graphene coating between the flake powder particles. Another reason is the recrystallization of adjacent matrix powder, which occurred during the sintering process. An obvious lamellar structure could, then, still be observed in the tensile fracture of the composite. In addition, there was no graphene pull-out phenomenon on the surface of the fracture, which was different from the graphene-reinforced copper matrix composites reported in the literature [11,12,15]. The main failure modes of graphene in composite materials are pull-out and fracture, and the failure modes are closely related to interface bonding, geometric factors, and inherent properties. The interface between graphene and copper is bonded by mechanical interlocking, and a large number of voids and cracks will be formed at the interface during tensile experiments, so the main failure mode of graphene is pull-out. After adding Cr, Ti, or other carbide-forming elements, the carbide particles at the interface can effectively transfer the load and prevent crack propagation in the tensile experiment, so the main failure mode of graphene is a fracture. What is more, the better structural integrity graphene has, the higher strength it owns, which makes the failure mode of graphene more prone to fracture [15,18,35,36]. The graphene material that has been used in the present study has higher structural integrity and higher tensile strength. Moreover, the carbide particles have precipitated to provide a strong interfacial bonding. Thus, the main failure behavior of graphene is the formation of fractures rather than pull-out. This is also the reason why the mechanical properties of the composites and the strengthening efficiency of graphene have been significantly improved.

## 4. Conclusions

By using Cu–Cr–Mg alloy powder and graphene nanosheets as raw materials, a GNPs/Cu–Cr–Mg composite has been prepared by using high-energy ball-milling and fast hot-pressing sintering. The microstructure of the composites, as well as the effect of the alloying elements and graphene on the mechanical properties and electrical conductivity of the composite, have been especially studied. The following results could be made:(1)Flake Cu–Cr–Mg powder was successfully prepared by controlling the parameters of ball-milling. The microstructure of the composite material was also a layered structure, and the directional distribution of graphene was realized. The Cr atoms were found to react with the amorphous carbon at the edge of graphene during the sintering process. Moreover, the Mg atoms were found to limit the size of the carbides by segregating to the surface of the carbide, which effectively improved the binding strength of the graphene/copper interface;(2)Graphene maintained sharp G peaks and smaller D peaks after ball-milling and sintering. The value of I_D_/I_G_ was 0.55, which indicates that the structure of graphene remains intact during the preparation process, and there are few defects;(3)The tensile strength of the composite reached a value of 349 MPa, which was 46% higher than that of the matrix. Moreover, the enhancement efficiency of graphene was 136. Furthermore, the conductivity of the composite became 81.5% IACS, which was only 1% IACS lower than that of the matrix. This can be attributed to the improvement of the graphene/copper interface by the Cr and Mg elements, as well as the high strength and high conductivity of graphene.

## Figures and Tables

**Figure 1 materials-15-06166-f001:**
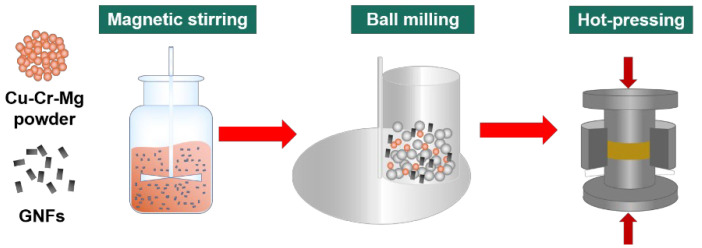
Preparation process of the GNPs /Cu–Cr–Mg composite.

**Figure 2 materials-15-06166-f002:**
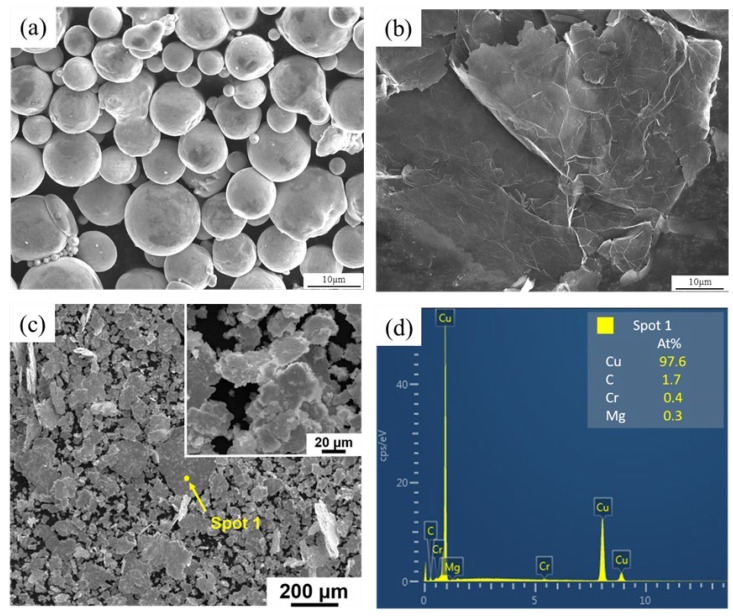
SEM images of GNPs/Cu–Cr–Mg composite powders; (**a**) gas atomized Cu–Cr–Mg alloy powder, (**b**) raw GNPs, (**c**) GNPs/Cu–Cr–Mg mixture powder, and (**d**) EDS results of spot 1.

**Figure 3 materials-15-06166-f003:**
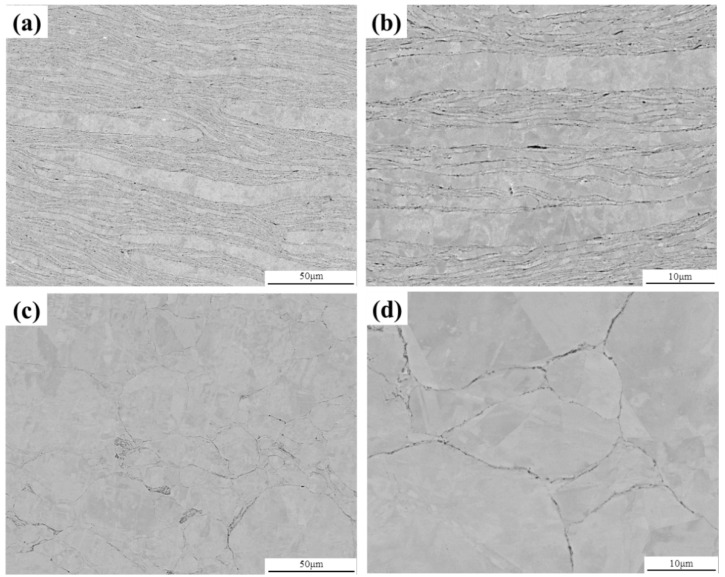
SEM images of a GNPs/Cu–Cr–Mg composite and the Cu–Cr–Mg matrix; (**a**) low magnification SEM image of the composite, (**b**) high magnification SEM image of the composite, (**c**) low magnification SEM image of the Cu–Cr–Mg matrix, and (**d**) high magnification SEM image of the Cu–Cr–Mg matrix.

**Figure 4 materials-15-06166-f004:**
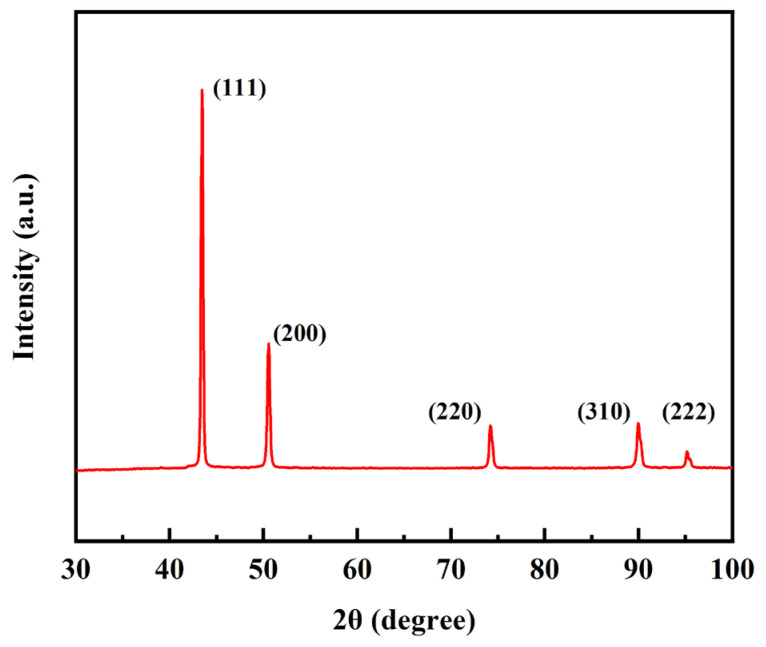
XRD patterns of a GNPs/Cu–Cr–Mg composite.

**Figure 5 materials-15-06166-f005:**
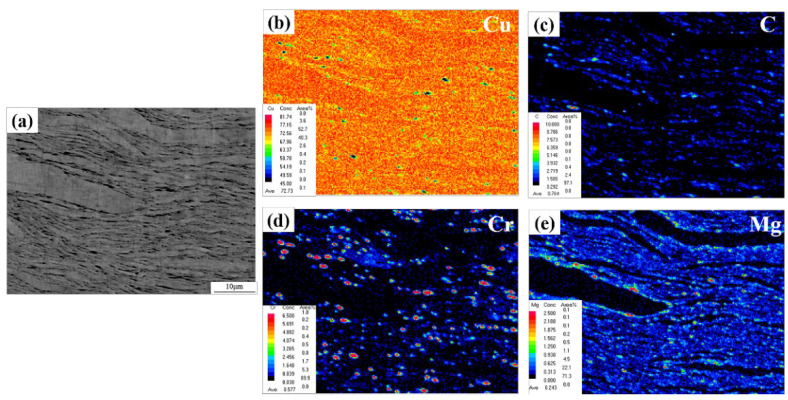
EPMA of a GNPs/Cu–Cr–Mg composite material; (**a**) SEM of a GNPs/Cu–Cr–Mg composite, (**b**) elemental mapping of Cu in (**a**), (**c**) elemental mapping of C in (**a**), (**d**) element mapping of Cr in (**a**) and, (**e**) elemental mapping of Mg in (**a**).

**Figure 6 materials-15-06166-f006:**
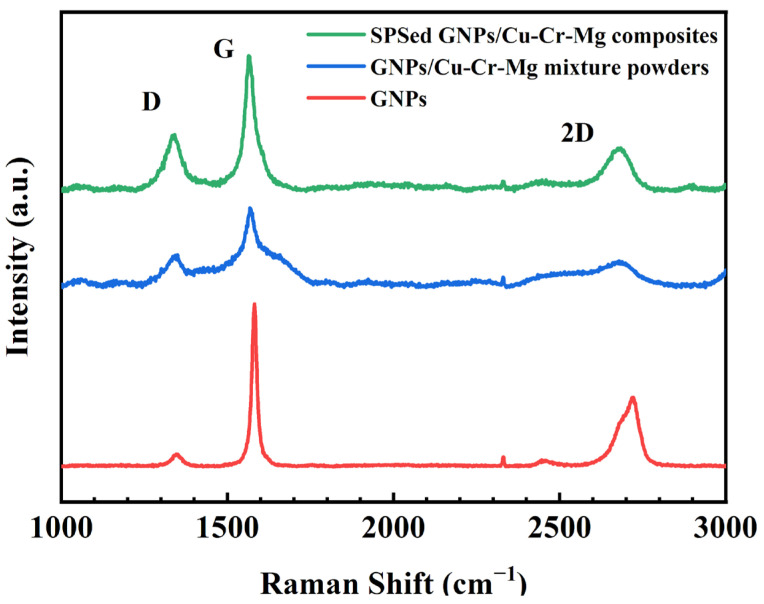
Raman spectrum of a GNPs/Cu–Cr–Mg composite.

**Figure 7 materials-15-06166-f007:**
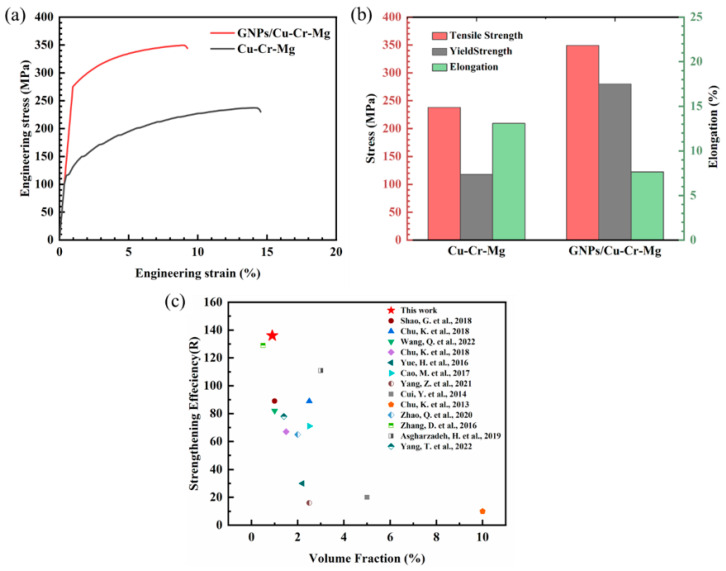
Mechanical properties of a GNPs/Cu–Cr–Mg composite; (**a**) tensile stress-strain curves, (**b**) obtained values of yield strength, tensile strength, and elongation of the composite and matrix material, and (**c**) the strengthening efficiency (R) obtained in this study in comparison with those reported in the literature [4,12,13,15,16,18,19,22,24,31,33,34,35].

**Figure 8 materials-15-06166-f008:**
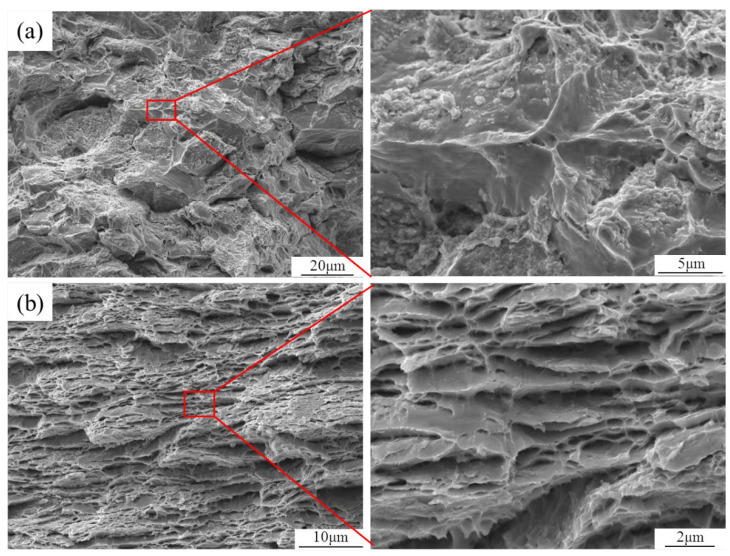
Tensile fracture of GNPs/Cu–Cr–Mg Composites. (**a**) Cu–Cr–Mg matrix material; (**b**) GNPs/Cu–Cr–Mg composites.

**Table 1 materials-15-06166-t001:** Chemical composition of the Cu–Cr–Mg alloy powder (at.%).

Element	Cu	Cr	Mg
Content	99.2	0.30	0.50

**Table 2 materials-15-06166-t002:** Relative density and electrical conductivity of the composites.

Element	Cu–Cr–Mg	GNPs/Cu–Cr–Mg
Relative density (%)	99.6	99.2
Electrical conductivity (%IACS)	82.4	81.5

## Data Availability

The data presented in this study are available on request from the corresponding author.

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
