# Peer review of "Microstructure and Properties of a Graphene Reinforced Cu–Cr–Mg Composite"

_materials, 2022, doi:10.3390/ma15176166_

Round 1

Reviewer 1 Report

Review report for materials-1878172

Manuscript ID: materials-1878172

Manuscript Title: Microstructure and Properties of a Graphene Reinforced Cu-Cr-Mg Composite

This manuscript explains about the Microstructure and Properties of a Graphene Reinforced Cu-Cr- Mg Composite. It is interesting and is good to the readers of the journal. It can be reconsidered after minor revision.

1.      Need re-written the abstract because it is mixed of past and present. It should be in one form.

2.        In table 1, the content of Cu, Cr and Mg are 99.2, 0.3 and 0.5 respectively. They should be same decimal in correct form such as 99.2, 0.30 and 0.50 respectively.

3.       In the text of the manuscript there are many decimal errors. Please make them corrections for the improvement of it.

4.       Need XRD of the composite.

5.      Need English corrections.

Author Response

Response to Reviewer 1 Comments

Dear Editor,

Thank you for your information of reviewing of our manuscript. We have revised the manuscript according to the reviewers’ comments line by line. The revised parts are also marked in red in the manuscript.

Point 1:Need re-written the abstract because it is mixed of past and present. It should be in one form.

Response 1: The abstract had been re-written carefully.

Point 2: In table 1, the content of Cu, Cr and Mg are 99.2, 0.3 and 0.5 respectively. They should be same decimal in correct form such as 99.2, 0.30 and 0.50 respectively.

Response 2: Decimal errors in table 1 had been corrected carefully.

Point 3: In the text of the manuscript there are many decimal errors. Please make them corrections for the improvement of it.

Response 3: Decimal errors in the text had been corrected carefully.

Point 4: Need XRD of the composite.

Response 4: The XRD of the composite had been detected. Related descriptions were added in Page: 5, Para: 2.

Point 5: Need English corrections

Response 5: The manuscript had been refined carefully.

Should you have any questions, please feel free to contact us.

Best regards,

Huichao Cheng

Reviewer 2 Report

The abstract of the article is written in general. I request the authors to emphasis the results in the abstract.

What is the novelty of the work. Typical application of the composition is not mentioned any where?

Some of the references cited in the article are nor relevant to the work and wheres as others are from same group of authors and country of origin.

The reference no 1 is cited after a big paragraph which can be avoided.

Processing of Mg in powder form is difficult in powder metallurgy. The experimental details are not well discussed here in the present format.

On what basis the authors chosen the Cr and Mg alloy. There is no much information available in manuscript.

Purity of GNF is not discussed much in the manuscript. The selected sintering temperature is very low in the study. The lanugage of technical content must be refine

Density discussion must include after the powder characteristics. The GNP as received powder morphology details not mentioned.

Fig 3 micro-structure is very poor. Please repeat the test. What is the driving force to do Raman analysis in this work

Most of the results were reported and not discussed technically. The authors must spend some time to discuss scientifically. Subset image can be placed in the place of low and high in fractography.

Major revision is required in the article

Author Response

Response to Reviewer 2 Comments

Dear Editor,

Thank you for your information of reviewing of our manuscript. We have revised the manuscript according to the reviewers’ comments line by line. The revised parts are also marked in red in the manuscript.

Point 1: The abstract of the article is written in general. I request the authors to emphasis the results in the abstract.

Response 1: The results had been emphasized in the abstract.

Point 2: What is the novelty of the work. Typical application of the composition is not mentioned any where?

Response 2: Firstly, the novelty of this work is that for the first time, both Cr and Mg elements are used to strengthen graphene/copper composites at the same time, which can improve the bonding strength of graphene/copper interface and enhance the strength of copper alloy matrix. This is unusual in previous reports, because the current matrix alloying method is limited to using only one element to improve the interface and does not consider strengthening the copper matrix. In this work, Cr is strengthened mainly by forming carbides at the interface, while Mg plays the role of solid solution strengthening and prevent the formation of excessive carbides on the interface. Secondly, graphene/copper composites have a wide range of applications in electrodes, rail transit, electronic packaging and electrical contacts.

Point 3: Some of the references cited in the article are nor relevant to the work and wheres as others are from same group of authors and country of origin.

Response 3: Some references nor relevant to the work had been deleted carefully. And some references related to this work are also replaced and supplemented. 

Point 4: The reference no 1 is cited after a big paragraph which can be avoided.

Response 4: On this issue, the first half of the introduction had been revised carefully. Related descriptions were added in Page: 1, Para: 1.

Point 5: Processing of Mg in powder form is difficult in powder metallurgy. The experimental details are not well discussed here in the present format.

Response 5: Relevant experimental details had been supplemented in the article. Related descriptions were added in Page: 2, Para: 4. Cu-Cr-Mg alloy powders were produced by gas atomization. The raw materials were high purity electrolytic copper (99.99 %), casting Cu-10Cr (wt.%) master alloy, and casting Cu-10Mg (wt.%) master alloy.

Point 6: On what basis the authors chosen the Cr and Mg alloy. There is no much information available in manuscript.

Response 6: The interface problem of graphene/copper composite can be improved by adding carbide forming elements. The addition of Cr element can form carbide particles at the interface, and the addition of Cr has little effect on the conductivity of the composite. Mg is a common solution strengthening element, and Mg does not react with graphene, which can play a better solid solution strengthening role and prevent the formation of excessive carbides on the interface. To sum up, Cr and Mg were selected as the additive elements.

Point 7: Purity of GNF is not discussed much in the manuscript. The selected sintering temperature is very low in the study. The lanugage of technical content must be refine.

Response 7: Firstly, graphene nanoplates in this study were purchased from XFNANO Materials CO., Ltd., Jiangsu, China. So I'm sorry that there were not much discussion about the purity of graphene. Secondly, I am sorry that due to my negligence, there are some errors in the description of sintering temperature in the manuscript. The composite was sintered at 900 ℃ and 50 MPa for 10 min, and related descriptions had been changed in Page: 3, Para: 2. Thirdly,the language of technical content has been refined carefully.

Point 8: Density discussion must include after the powder characteristics. The GNP as received powder morphology details not mentioned.

Response 8: Density discussion have been supplemented in the article. Related descriptions were added in Page: 7, Para: 2. The morphology of GNPs had been added in the article and related descriptions were added in Page: 2, Para: 4.

Point 9: Fig 3 micro-structure is very poor. Please repeat the test. What is the driving force to do Raman analysis in this work

Response 9: Firstly, the text of Fig 3 has been repeated carefully. Secondly, graphene is a two-dimensional atomic crystal structure formed by the tight packing of sp2 carbon atoms. The integrity of its structure is an important factor to ensure the electrical conductivity and enhance the ability of graphene. The destruction of graphene structure during the preparation process can be observed intuitively by Raman spectroscopy.

Point 10: Most of the results were reported and not discussed technically. The authors must spend some time to discuss scientifically. Subset image can be placed in the place of low and high in fractography.

Response 10: On this issue, we have more scientific discussions in the results and discussions, especially in Page: 7, Para: 1 and 2, Page: 8, Para: 4. And subset image also had be placed in the place of low and high in fractography in Fig 8.

Should you have any questions, please feel free to contact us.

Best regards,

Huichao Cheng

Round 2

Reviewer 2 Report

The authors have made the tremendous changes in the manuscript and I am satisfactorily with the answers they provided during the review process.